# 3,6′-Dithiopomalidomide Ameliorates Hippocampal Neurodegeneration, Microgliosis and Astrogliosis and Improves Cognitive Behaviors in Rats with a Moderate Traumatic Brain Injury

**DOI:** 10.3390/ijms22158276

**Published:** 2021-07-31

**Authors:** Pen-Sen Huang, Ping-Yen Tsai, Ling-Yu Yang, Daniela Lecca, Weiming Luo, Dong Seok Kim, Barry J. Hoffer, Yung-Hsiao Chiang, Nigel H. Greig, Jia-Yi Wang

**Affiliations:** 1Graduate Institute of Medical Sciences, College of Medicine, Taipei Medical University, Taipei 110, Taiwan; benson342536@gmail.com (P.-S.H.); murphy7160@gmail.com (P.-Y.T.); allison73323@gmail.com (L.-Y.Y.); 2School of Medicine, College of Medicine, Taipei Medical University, Taipei 110, Taiwan; 3Drug Design & Development Section, Translational Gerontology Branch, Intramural Research Program, National Institute on Aging, NIH, Baltimore, MD 21224, USA; daniela.lecca@nih.gov (D.L.); luowe@grc.nia.nih.gov (W.L.); 4Aevis Bio, Inc., Daejeon 34141, Korea; dskim@aevisbio.com; 5Aevis Bio, Inc., Gaithersburg, MD 20878, USA; 6Department of Neurological Surgery, Case Western Reserve University, Cleveland, OH 44106, USA; bjh82@case.edu; 7Department of Neurosurgery, Taipei Medical University Hospital, Taipei Medical University, Taipei 110, Taiwan; ychiang@tmu.edu.tw; 8Neuroscience Research Center, Taipei Medical University, Taipei 110, Taiwan; 9Department of Surgery, School of Medicine, College of Medicine, Taipei Medical University, Taipei 110, Taiwan

**Keywords:** 3,6′-dithiopomalidomide, pomalidomide, traumatic brain injury, neurodegeneration, cognitive deficits, neuroinflammation, microgliosis, astrogliosis, immunomodulatory imide drugs

## Abstract

Traumatic brain injury (TBI) is a leading cause of disability and mortality worldwide. It can instigate immediate cell death, followed by a time-dependent secondary injury that results from disproportionate microglial and astrocyte activation, excessive inflammation and oxidative stress in brain tissue, culminating in both short- and long-term cognitive dysfunction and behavioral deficits. Within the brain, the hippocampus is particularly vulnerable to a TBI. We studied a new pomalidomide (Pom) analog, namely, 3,6′-dithioPom (DP), and Pom as immunomodulatory imide drugs (IMiD) for mitigating TBI-induced hippocampal neurodegeneration, microgliosis, astrogliosis and behavioral impairments in a controlled cortical impact (CCI) model of TBI in rats. Both agents were administered as a single intravenous dose (0.5 mg/kg) at 5 h post injury so that the efficacies could be compared. Pom and DP significantly reduced the contusion volume evaluated at 24 h and 7 days post injury. Both agents ameliorated short-term memory deficits and anxiety behavior at 7 days after a TBI. The number of degenerating neurons in the CA1 and dentate gyrus (DG) regions of the hippocampus after a TBI was reduced by Pom and DP. DP, but not Pom, significantly attenuated the TBI-induced microgliosis and DP was more efficacious than Pom at attenuating the TBI-induced astrogliosis in CA1 and DG at 7D after a TBI. In summary, a single intravenous injection of Pom or DP, given 5 h post TBI, significantly reduced hippocampal neurodegeneration and prevented cognitive deficits with a concomitant attenuation of the neuroinflammation in the hippocampus.

## 1. Introduction

Traumatic brain injury (TBI) is a leading cause of injury-related death and long-term disability, with an estimated 69 million people suffering a TBI annually worldwide [1]. Some 4.8 million cases occur in North America. The preponderance of TBIs are mild to moderate in nature, representing 80–95% of cases, with severe TBI compromising the remaining 5–20% [2,3]. With improved survival following the original injury, TBI may elicit significant and lifelong cognitive and behavioral deficits and physical impairments that necessitate long-term care [2,3,4]. Indeed, TBI represents a progressive process rather than a single event. It is one of the most important risk factors for the subsequent development of neurodegenerative disorders, particularly Parkinson’s disease and Alzheimer’s dementia [2,5], and is associated with changes in gene expressions in pathways that lead to these disorders [6,7]. The increasing incidence of TBI represents one of the largest global burdens of disease [8]. 

TBI-mediated brain damage is routinely divided into two phases. At the moment of injury, initial damage that often manifests as a contusion or laceration and generally involves diffuse axonal injury and intracranial bleeding result in immediate (necrotic) cell death [9], which is largely considered irreversible. This is followed by a prolonged and progressive secondary phase involving pathophysiological cascades that are associated with neuroinflammation, ischemia and glutamate excitotoxicity that lead to microglial and astrocyte reactivity and, eventually, to neural cell dysfunction and death [10,11]. These secondary processes provide mechanisms for a possible drug intervention [12]. 

The instigation of an inflammatory response following a TBI is considered crucial for triggering neuronal-reparative mechanisms [11,13,14,15]. However, if disproportionate and not tightly controlled, these same processes can drive neuronal dysfunction and neurodegeneration by causing a self-propagating neuroinflammatory cascade [16]. Consequent to a TBI, there is a significant elevation in microglia and astrocyte activation, altering their phenotype and role from a physiological, surveillant one that supports and optimizes neuronal function to a more pathologically orientated one involved in wound stabilization and repair that requires a delicate balance between the production and release of pro- and anti-inflammatory cytokines and chemokines [17,18].

The hippocampus is critical in explicit memory formation and, as part of the limbic system, is implicated in mood disorders, including major depression and anxiety [19]. Being particularly vulnerable to TBI, the fornix and hippocampus volumes decline in correlation with injury severity, and memory performance is associated with the residual hippocampus volume [20]. In addition, a significant reduction in long-term potentiation (LTP) was reported for the ipsilateral CA1 and dentate gyrus (DG) regions of the hippocampus from 1 to 7 days following a mild TBI [21]. Multiple studies across TBI animal models have shown short- and long-term learning and memory dysfunction [12], and many have involved hippocampal injury. Increasing evidence indicates that post-TBI dissemination of brain inflammation extends beyond the peri-lesion site to remote brain regions [22]. Excessively activated microglia and astrocytes in the ipsilateral and contralateral hippocampus can persist for more than a week after a TBI and drive neurodegenerative processes that include impaired neurogenesis and reparative mechanisms [23]. In large part, neuroinflammatory processes across the brain, and particularly in the hippocampus, are driven by TNF-α-responsive genes [24]. 

The current study characterized the therapeutic potential of a novel pomalidomide (Pom) agent, namely, 3,6′-dithiopomalidomide (DP) (Figure 1), in comparison with Pom in the hippocampus following a TBI induced by a controlled cortical impact (CCI) in rats. DP and Pom are members of the immunomodulatory imide drug (IMiD) class, which lowers TNF-α synthesis and hence downstream signaling [25]. Our previous study demonstrated that post-injury treatment with pomalidomide (Pom) [26] or 3,6′-dithiopomalidomide (DP) [25], a novel analog of Pom, can alleviate neuronal loss and suppress TBI-induced neuroinflammation in the cerebral cortex by 24 h after a TBI. In addition, DP can improve sensorimotor behavior outcomes following a TBI [25]. In this present study, we investigated the effectiveness of Pom and DP on hippocampal injury, neuroinflammation and neurodegeneration 7 days after a TBI. Furthermore, to correlate our findings with clinical efficacy, we evaluated whether Pom and DP could improve TBI-induced cognitive deficits, including short-term memory impairment and anxiety-like behavior.

## 2. Results

### 2.1. Post-Injury Treatment with Pom and DP Reduced Hippocampal Injury Caused by TBI with DP Proving More Efficacious in Reducing Injury Volume Than Pom at 24 h and 7 Days (7D) after TBI

We used cresyl violet to stain the Nissl substance in the cytoplasm of neurons. The TBI resulted in a loss of neurons in the hippocampal region as reflected by gross reductions in cresyl violet staining intensity in the ipsilateral brain at 24 h and 7D after a TBI as compared with sham animals. In contrast, the cytoarchitecture in the contralateral hemisphere remained normal. The contusion volume (tissue loss) in the TBI + Veh animals was in the order of 16.2 ± 4.3% of the contralateral hemisphere volume at 24 h and 21.8 ± 2.6% at 7D after a TBI (Figure 2A). Both DP and Pom reduced the contusion volume due to the TBI by 24 h and 7D, but DP was more efficacious regarding the volume reduction than Pom by both 24 h and 7D after a TBI (Figure 2B).

### 2.2. DP Exhibited an Earlier Onset of Improvement in TBI-Induced Anxiety-Like Behavior Than Pom within 24 h after TBI; the Effects of DP and Pom Were Comparable at 7D after TBI

The open-field test (OFT) was used to evaluate both general locomotion and anxiety-like behaviors (Figure 3). In the traces and heatmaps for the OFT, the rats in the TBI + Veh group spent most of their time in the corner without moving to the central area, reflecting the anxious behavior of the animals. This phenomenon began as early as 24 h and persisted for up to the 7D after the TBI. DP, but not Pom, reversed the TBI-induced anxiety by 24 h (Figure 3A,B). The mean distance, which represented the locomotion of the rats, significantly decreased at 24 h and was not restored at 7D after a TBI. The restoration of locomotion to pre-TBI levels was seen in the TBI + DP group by 24 h, and in the TBI + Pom group by 7D after a TBI (Figure 3C). The results are in congruence with the more efficacious effect of DP on the reduction of contusion volume at 24 h.

### 2.3. Short-Term Memory Impairment Was Significant at 7D after TBI and, although Both DP and Pom Improved Short-Term Memory, Only DP Fully Restored It at 7D after TBI to Pre-TBI Levels

Short-term memory was evaluated using the novel object recognition (NOR) test at 24 h and 7 days after a TBI. The NOR, which is quantified using the discrimination index, tests the animals’ exploratory behavior, which is based on short-term memory. The time spent on the novel object was longer in sham groups (i.e., greater discrimination index) because their short-term memory was intact (Figure 4A,B). The significantly lower discrimination index, which indicated impaired short-term memory, was seen at 7D following a TBI. The DP and Pom groups showed short-term memory improvement when compared to the TBI + Veh group. However, Pom, unlike DP, failed to completely restore short-term memory, as evidenced by a decreased discrimination index in the Pom group at 7D after a TBI compared to the pre-TBI condition (Figure 4C).

### 2.4. Pom and DP Reduced TBI-Induced Neurodegeneration in the Hippocampus at 7D after TBI but DP Reduced Numbers of Degenerating Neurons More Than Pom in the DG

To visualize degenerating neurons, FJC staining was performed. TBI induced neurodegeneration within the contusion region in the CA1 and DG regions in the hippocampus (Figure 5A). FJC-positive cells with a neuronal morphology were evident at 24 h after a TBI in the CA1 and DG regions (Figure 5D). Notably, there was a significant decrease in the number of FJC-positive cells in both the TBI + Pom and TBI + DP groups in the CA1 region (Figure 5B,C). However, the DP treatment produced a statistically significant decrease in the number of FJC-positive cells in the DG region compared to the TBI + Pom group (Figure 5C).

### 2.5. DP Reduced TBI-Induced Microgliosis in the Hippocampus at 7D after TBI

Microglial numbers, which were measured using Iba1 immunocytochemistry, were significantly elevated in the CA1 and DG regions 7D after a TBI in the hippocampus with a greater increase in the DG (Figure 6A). Only DP significantly reduced Iba1+ cells in both the CA1 and DG at 7D after a TBI. Pom reduced Iba1+ cells in the CA1, but not the DG. In addition, DP was more efficacious in suppressing microgliosis than Pom in both the CA1 and DG (Figure 6B,C).

### 2.6. DP Was More Effective Than Pom at Attenuating TBI-Induced Astrogliosis in the Hippocampus

The number of astrocytes, labeled using GFAP, was significantly increased after a TBI at 7D in the CA1 and DG regions in the hippocampus (Figure 7A). Pom and DP significantly reduced the GFAP+ cells in the CA1 and DG fields at 7D after a TBI, with DP exhibiting a better efficacy than Pom (Figure 7B,C).

## 3. Discussion

Numerous recent studies indicate that inflammation is a cardinal homeostatic reaction to the damage that occurs following a TBI [27,28]. Though initiated to trigger reparative mechanisms, an over-zealous and protracted response can be detrimental and drive neurological dysfunction. As such, regulating neuroinflammation represents a prospective therapeutic strategy for TBI, providing numerous potential drug targets. There are approximately 4.8 million cases of TBI annually in North America alone [1]. Unfortunately, despite over 30 controlled clinical trials on TBI since 1993, there are no currently approved pharmacological treatments for this disorder [29]. We synthesized DP, which is a novel 3,6′ thionated analog of Pom, as a new IMiD that potently lowers TNF-α synthesis but still allows it to undergo a time-dependent lower release following a pathological insult and, thereby, quells any potentially excessive inflammatory response [25]. In our previous study, DP significantly and dose-dependently reduced cerebral cortex contusion volume and improved sensory-motor functional outcomes at 24 h [25], with a therapeutic window of between 5 and 7 h for a CCI injury in a well-characterized rodent model of moderate TBI. Biochemical and immunohistochemical investigations of the cortical contusion tissue demonstrated that DP mitigates TBI-induced neuronal loss via apoptosis and autophagy and attenuates microglial activation and astrogliosis in an injured cerebral cortex at 24 h after a TBI. Notably, DP exhibited greater efficacy than Pom. In the present study, not only did we demonstrate a further increased contusion volume at 7D (21.8 ± 2.6%) compared to 24 h (16.2 ± 4.3%) after a TBI but we also demonstrated that a single intravenous injection of Pom or DP could effectively reduce the even larger contusion volume (Figure 2). The current study extended this initial research to document the longer-term efficacy of DP and, to a lesser extent, Pom over 1 to 7 days and, importantly, focused on the hippocampus. This brain region is particularly vulnerable to a TBI, is fundamental in memory acquisition, and is implicated in mood disorders that include anxiety and depression [19,20]. 

Neuroinflammation clearly plays a dual role in TBI; when appropriately and time-dependently controlled, it can augment the removal of cellular debris and instigate neuro-regenerative processes post injury. However, if excessive, neuroinflammation can also trigger neuronal cell death and neurodegeneration [28]. Although classical anti-inflammatory treatments have demonstrated favorable outcomes in TBI preclinical models, such findings have not translated into positive randomized placebo-controlled human clinical trials. Indeed, in some cases, anti-inflammatory treatments may be detrimental [30,31]. In this light, the IMiD drug class was chosen consequent to its known action to lower TNF-α generation and proinflammatory cytokine levels [32], as a new anti-inflammatory strategy to mitigate TBI.

The initiation and promotion of the innate immune response following a head injury involve an intricate choreography of many factors. The lysis of cells during the initial stage of brain injury permits the release of damage-associated molecular patterns (DAMPs). These trigger a fast and dramatic elevation in TNF-α generation and release. Systemic and brain levels of TNF-α rise acutely after injury, preceding the ensuing elevation of other inflammatory cytokines, and subsequently decline after about 24 h [33,34]. Such a rise in TNF-α can promote glial activation, monocyte infiltration and neuronal loss and can reduce BBB integrity and neurogenesis [35]. Moreover, TNF-α receptor-mediated activation of NF-ĸB, which is a central mediator of proinflammatory gene induction, can transcriptionally induce further TNF-α and amplify TNF-α signaling pathways. These increase the expression of multiple other cytokines, chemokines and their receptors, and elicit the often poorly regulated immune response that may underlie the acute, subacute and subsequent chronic neuroinflammation that invariably follows brain injury [36,37]. 

Pre-clinical studies of etanercept, which is a fusion protein that amalgamates the TNF receptor to the constant end of the IgG1 antibody and, thereby, lowers TNF-α by acting as a decoy receptor, support the targeting of TNF-α to potentially mitigate TBI [37]. The administration of etanercept in rats with fluid-percussion-induced brain injury quelled microglial cell activation, lowered TNF-α generation and, by this means, mitigated both motor and neurological deficits at 3 days post injury [38]. Likewise, open-label human TBI etanercept studies demonstrated favorable results [39]. Thalidomide increases the mRNA degradation of TNF-α via post-transcriptional mechanisms through elements within its 3′-untranslated region [40] and thus lowers TNF-α generation [41]. Unfortunately, the use of thalidomide in animal disease models and humans has proved difficult to translates as an approach to reduce neuroinflammation accompanying neurodegeneration [42] consequent to the development of intolerable off-target actions that proved dose-limiting prior to clinically relevant TNF-α lowering anti-inflammatory effects, as evidenced in a recent human Alzheimer’s disease thalidomide clinical trial [43]. This prompted a recently initiated lenalidomide Alzheimer’s disease clinical trial (MCLENA-1) [44], as lenalidomide is a more potent second-generation analog against TNF-α within the IMiD drug class [45]. Additionally, there is well-documented teratogenic sequela with this drug class that, although not potentially relevant for Alzheimer’s patients due to their age, may be relevant for TBI subjects, thereby motivating the search for new compounds that are less encumbered by prohibitive adverse actions [32,41,46]. Replacing the oxygen of select carbonyl groups with sulfur within thalidomide, particularly in the 3 and 6′ positions (Figure 1), augments the TNF-α-lowering action [47]. Furthermore, the addition of an amino group to the fourth carbon of the phthaloyl ring to yield Pom as a third-generation thalidomide analog likewise increases the TNF-α potency [48]. The application of both structural changes resulted in DP.

DP and Pom, which are similar to thalidomide but unlike TNF-α-lowering biologicals, both freely enter the brain and achieve a brain-to-plasma concentration-ratio of 0.8 [25]. Given systemically as a single dose, DP and Pom capably lower both systemic and brain elevations in TNF-α levels that are induced by a classical bacterial lipopolysaccharide challenge and significantly reduced TBI-induced secondary phase damage when administered within 5 h post injury. On evaluating the dose dependence, DP demonstrated greater potency than Pom, as shown previously in our acute study [25].

Evaluation of behavioral parameters was undertaken in the current study to further define DP and Pom’s actions over an extended time course. Although the mean overall distance traveled (locomotion) (Figure 3) was decreased after a TBI, similar to our previous findings [49], some studies showed increases in locomotion in severe TBIs [50,51]. Several tests for anxiety-like behaviors were proposed, including open-field, elevated zero maze and elevated plus maze tests, but it should be noted that the results are inconsistent between tests. The locomotion can be confounded by impaired motor function. Furthermore, the level of injury severity, different models and even gender lead to discrepant results [52]. Thus, tests with a higher sensitivity/specificity, as well as relevant mechanisms for post-TBI anxiety, should be further investigated. Consistent with our previous results [49], the discrimination index evaluated using NOR (Figure 4) in the present study decreased from 60–70% to 50% at 7 days post TBI. Studies demonstrated the index further declined to 27% at 14 days after a moderate TBI [53], and a significantly lower discrimination index persisted up to 8 weeks after repetitive mild TBIs [54]. Interestingly, a TBI contributes to a persistent cognitive inflexibility at 9 months after the TBI, which is associated with atrophy in several brain structures, including the hippocampus [55]. These results confirmed the long-term deficits following a TBI and the importance of early intervention. The quantification of FJC-positive degenerating cells close to the injury site supports the favorable potencies of DP and Pom, with the former demonstrating the greater activity (Figure 5). This correlated with alterations in GFAP reflecting injury-related gliosis. When cellular damage becomes appropriately intense, neuronal cell death ensues, which is instigated by one or more of multiple biochemical cascades [56]. The resulting degree and distribution of cell death clearly depend on the TBI severity, injury site and age of the host, but additionally appears to relate to the brain area evaluated and proximity to the injury site [57,58,59]. The hippocampus is considered particularly vulnerable across TBI animal models, especially the dentate gyrus, where neural stem cells reside that can generate new neurons throughout the life cycle [59,60]. Neuronal cell death appears to peak after a moderate TBI at approximately 24 h and persists for up to 14 days [59]. Our study results were in line with this, as FJC-positive cells were elevated to almost 500/mm^2^ at 24 h post TBI in our early study [25], as compared to approximately 50/mm^2^ at 7 days in the current study. Within the hippocampal dentate gyrus, immature (i.e., newborn) granular neurons that express the neural cell adhesion molecule (NCAM) but not the mature neuronal marker NeuN appear to be particularly vulnerable to injury [60]. The majority of degenerative neurons after a TBI appear to be distributed in the DG rather than the CA1 or CA3 regions [61], which is in line with the results from our present study. Specifically, the number of FJC+ neurons determined in DG was twice as many as those in the CA1 region (Figure 5). Clearly, such immature neuron loss can compromise hippocampal neurogenesis [60], particularly when associated with neuroinflammation [23,43], which can contribute to the memory and learning impairments that occur following a TBI. 

Iba1 is a widely used pan-microglial/macrophage marker and its expression level increases in line with microglial activation [62]. TBI induces a change in microglial phenotype to an activated (simplistically defined as M1) state from a quiescent surveillant state [36,63], and this is accompanied by a dramatic elevation in proinflammatory protein expression [25]. Such effects were substantially mitigated using DP, and to a lesser degree by Pom, with DP demonstrating greater potency in our earlier cerebral cortex evaluation [25]. Clinical studies showed that amounts of Iba-1-immunopositive microglia and GFAP-immunopositive astrocytes were found to increase significantly in brain samples of TBI patients [64,65]. The Iba1+ cells began to increase after 1 day, reached the plateau at 3 days and persisted for a month after a TBI in the human cortex [64]. Caplan et al. found significantly increased Iba1+ cells (250/mm^2^) in the hippocampus at 28 days, not 24 h, after a TBI [66]. In this study, the number of microglia increased in both the CA1 (350/mm^2^) and DG (650/mm^2^), which indicated the microgliosis in the hippocampus occurred as early as 7 days after a TBI.

GFAP, which is a monomeric intermediate filament protein that is concentrated in the cytoskeleton of astrocytes and is specific to brain tissue, is a well-characterized marker for mature and differentiated brain astrocytes [67]. Following a TBI, brain edema is found with vasogenic edema occurring rapidly after the injury, primarily in the center of the lesion, whereas cytotoxic edema has a later onset and is predominant [68]. There is growing evidence that family members of aquaporins play an important role in traumatic brain injury edema. Using double immunofluorescence staining of aquaporin-4 (AQP4) and GFAP, we observed increased co-localization of AQP4 and GFAP in the hippocampus of TBI-affected animals (data not shown). In our GFAP staining results, in addition to increased numbers of astrocytes (GFAP-positive cells) in the TBI animals, there was also morphological evidence of hypertrophy or swelling of these cells (Figure 7D), suggesting the formation of TBI-induced edema in astrocytes. In our current study, DP and, to a lesser extent, Pom mitigated TBI-induced astrogliosis, which, like microglial activation, was more prominent in the DG after injury (Figure 7).

In our previous study [25], we demonstrated the molecular mechanism of DP (lowering of TNF-α generation in plasma, the hippocampus and the cerebral cortex, as well as neuroprotection against apoptotic and autophagic death). In addition to their effects on reducing astrogliosis and microglia activation, Pom and DP also downregulated inflammatory proteins (iNOS and COX-2) [25]. Our results in this study showed that cognitive behaviors were significantly impaired at 7D post TBI, suggesting functional deficits within the hippocampus. Several studies demonstrated a synaptic loss in the hippocampal CA1 region without neuronal death at 7 days after a TBI, which was correlated with cognitive dysfunction. For example, the expression of PSD-95 (postsynaptic density-95), which is a post-synaptic protein, decreased in the hippocampus as early as 4–7 days after a TBI, with the loss of PSD-95 being directly correlated with a reduction in cognitive function [69,70]. Reduced PSD-95 expression in the hippocampus paralleled impaired long-term potentiation (LTP), which is a well-accepted electrophysiological index of synaptic strength that underlies learning and memory [70]. Similar reductions of other synaptic markers (Glu R1 or synaptophysin) were also reported [71]. In addition, a clinical study found evidence of dendritic swelling and synaptic phagocytosis by astrocytes and microglia in TBI patients at the EM level [72]. It is likely that a local inflammatory synaptic loss, as well as neurodegeneration, might contribute to behavioral defects. Although the severity and duration of TBI may result in various synaptic changes [73], we did intend to further examine the synaptic loss at 1 month in our future study based on the reported phagocytic activity of microglia and astrocytes to synapses at 1 month [61,74].

Taken together, our results show progressive increases in TBI-induced contusion volume, paralleled by increases in neurodegeneration, microgliosis and astrogliosis within the hippocampus. In all cases, DP and, to a lesser degree, Pom reduced these changes at 24 h and 7 days, which might be associated with favorable changes in anxiety-like locomotor behavior, and also reduced short-term memory deficits induced by TBI. Our animal studies with DP and Pom support the notion that these drugs warrant further development and evaluation in additional even-longer-term follow-up studies after a TBI, and may additionally be valuable in other neurological disorders that involve a neuroinflammatory response. These excess astrocytes and microglia may contribute to synaptic loss, leading to cognitive deficits, and a potential recovery induced by Pom and DP treatments warrants long-term evaluation.

## 4. Conclusions

In this current study, the main focus was to highlight the fact that a focal cortical injury due to a CCI induces diffuse, widespread hippocampal neurodegeneration at 7D post TBI and impaired cognitive behaviors. Importantly, a single systemic intravenous dose of DP or Pom administered at 5 h post injury significantly reduced hippocampal neurodegeneration and improved cognitive behaviors at 7D after a TBI, paralleled with a significant reduction in astrogliosis and microgliosis. Our animal studies with DP and Pom support the notion that these drugs warrant further development and evaluation in additional even-longer-term follow-up studies in TBI, and may additionally be valuable in other neurological disorders that involve a neuroinflammatory response. 

## 5. Materials and Methods

### 5.1. Animal Model of TBI

An animal model of a moderate TBI induced by a CCI was used to evaluate brain injury, as previously detailed [49,75,76,77,78] and briefly described below. Male Sprague Dawley (SD) rats (weighing 250–300 g) were anesthetized and then placed in a stereotaxic frame. A 5 mm craniotomy was performed over the left parietal cortex, centered on the coronal suture and 3.5 mm lateral to the sagittal suture. The rat CCI model used an electromagnetic impactor device that possessed a rounded (5 mm diameter) at a velocity of 4 m/s to a depth of 2 mm below the dura, which resulted in an injury of moderate severity. Sham animals received a craniotomy but were not subject to a CCI injury. Subsequent to the sham or TBI procedures, the rats were randomly allocated to treatment or Veh groups. The number of rats selected per group is detailed within each figure legend and is based on our previous studies [25,49,77,78] and a power analysis (α = *p* < 0.05 and 1 − β = 0.08). Behavioral evaluations were conducted prior to the TBI or sham procedures and at 24 h and 7 days subsequent to each TBI. Rats were subsequently euthanized to support the histological and biological analyses.

### 5.2. Drug Synthesis and Administration

POM (4-amino-2-(2,6-dioxopiperidin-3-yl)-isoindole-1,3-dione) was generated via a two-step synthetic procedure. Initially, 3-aminopiperidine-2,6-dione was condensed with 3-nitrophthalic anhydride in refluxing acetic acid. Subsequent precipitation over ice water (0 °C) provided the consequent nitro-thalidomide as a grey-purple solid. Later hydrogenation over a palladium catalyst generated Pom as a yellow solid and chemical characterization was used to confirm its structure. Part of this Pom was then used as starting material for DP synthesis via selective thionation in the 3 and 6′ positions and chemical characterization was used to confirm its structure again. For the TBI studies, DP and Pom were freshly prepared prior to administration at a dose of 0.5 mg/kg in 100% DMSO. Immediately prior to use, they were diluted with phosphate-buffered saline (PBS) to 10% DMSO and injected intravenously (i.v.) at 5 h post TBI. Care was taken to ensure that the drugs remained in solution when diluted with PBS. Animals receiving the vehicle (Veh) were administered 10% DMSO in PBS (volume: 1 mL/kg).

### 5.3. Behavioral Evaluation of Neurological Outcomes

Behavior testing was performed at 24 h and 7 days after the TBI by an observer who was blind to the experimental treatments. The assessments consisted of a novel object recognition (NOR) test and an open-field test (OFT). These procedures were performed as previously described [25,26,49].

### 5.4. Open-Field Test (OFT)

OFT was used to measure general locomotor activity and anxiety-like behavior as previously described [49,79]. Before the TBI induction, the animals were trained for 3 days. At 24 h and 7 days after the TBI, the animals were put in an open-field apparatus (60 × 60 × 100 cm in dimension) and a video camera was equipped above the apparatus to record each 10-min trial. The mean overall distance traveled and the heatmaps of each animal were recorded and analyzed using the EthoVision XT 11.0 tracking system.

### 5.5. Novel Object Recognition Test (NOR)

The NOR test was used to assess hippocampal-mediated recognition memory and was conducted as we previously described [49]. The NOR apparatus consisted of an open field (60 × 60 × 100 cm in size) with 2 adjacently located imaginary circular zones and a video camera fixed on the top wall of the apparatus. There are three phases in the NOR process: habituation, familiarization and a test phase. Briefly, animals were allowed 10 min to individually habituate themselves to the open-field box for 3 days in the habituation phase. After 3 days, two identical objects (A + A) were adjacently placed in the apparatus and the animals were allowed to explore for a 10 min familiarization phase. After 1 h, a test trial was then performed by replacing a familiar training object with a novel object (A + B) and allowing the rats to explore for 10 min. The duration of time spent with each object was recorded and analyzed using the EthoVision XT 11.0 tracking system. The discrimination preference index was calculated. The discrimination preference index = [(time (in seconds) spent exploring the novel object − time spent exploring the familiar object)/(total time (in seconds) spent exploring both objects)] × 100%.

### 5.6. Contusion Volume

Briefly, brain sections were stained with cresyl violet to determine the volume of TBI-induced injury in the ipsilateral hemisphere 24 h or 7 days following a TBI. Stained sections were digitized and quantified using Image J (National Institutes of Health, Bethesda, MD). The volume of the injury areas was computed by adding together the number of slices and multiplying this by the inter-slice distance (500 μm). The percent of hemisphere tissue loss was calculated with the following formula: [(contralateral hemispheric volume − ipsilateral hemispheric volume)/(contralateral hemispheric volume) × 100%], as previously described [80] and widely used [25,26,49,77,78].

### 5.7. Fluoro-Jade C (FJC) Staining

To visualize degenerating neurons, the FJC Ready-to-Dilute Staining Kit (Biosensis, TR-100-FJ, Thebarton, South Australia) was used according to the manufacturer’s instructions with some modifications to detect neurodegeneration in the hippocampal tissue [26,49,81]. Briefly, slides were dewaxed in xylene and rehydrated in a graded ethanol series, followed by a rinse in distilled water for 3 min. The slides were then incubated in potassium permanganate (1:15) for 10 min to quench the non-specific staining. Then, the slides were rinsed again in distilled water for 2 min, and subsequently in Fluoro-Jade C (1:25) and DAPI solution for 15 min in the dark. Following incubation, the slides were then washed in distilled water and then dried at 50–60 °C in an oven. All sections were observed and photographed under a fluorescence microscope with a blue (450–490 nm) excitation light. The images were opened with the SPOT image analysis software (Diagnostic Instruments, Sterling Heights, MI, USA) and FJC-positive cells in three randomly selected fields were quantified using ImageJ (NIH, Bethesda, MD, USA).

### 5.8. Immunohistochemistry (IHC)

For the immunohistochemical staining, paraffin-embedded brain sections (10 µm) from various treatment groups were dewaxed in xylene and rehydrated through a graded ethanol series, followed by a rinse in PBS for 3 min. Antigen was retrieved via a heat treatment with 0.1% citrate buffer (pH 6.0) for 15 min. Sections were then permeabilized and blocked for 60 min in 5% BSA and 0.2% Triton X-100. After blocking, sections were subsequently incubated with primary antibody overnight at 4 °C. The rabbit polyclonal anti-Iba1 (GeneTex, GTX100064, 1:200) or rabbit polyclonal anti-GFAP (GeneTex, GTX108711, 1:400) were used as the primary antibodies. Following overnight incubation, sections were washed in PBS and incubated with VECTASTAIN Elite ABC HRP Kit (Peroxidase, Rabbit IgG and Mouse IgG) for an hour at room temperature. All images were acquired using a microscope (IX70, Olympus, Tokyo, Japan) attached to a digital camera and SPOT imaging software. Quantification of the numbers of GFAP- and Iba-1-positive cells in the hippocampus was performed with Image J software as we described previously [25].

### 5.9. Statistical Analysis

The bar graphs are expressed as mean ± standard error of the mean (SEM). Comparisons between multiple groups were made using one-way analysis of variance (ANOVA) followed by Tukey’s or Dunnett’s tests using SigmaPlot 10 and SigmaStat 3.5. (Jandel Scientific Corp., San Rafael, CA, USA). A probability level of *p* < 0.05 or less was considered statistically significant and specific values are noted in figure legends.

## Figures and Tables

**Figure 1 ijms-22-08276-f001:**
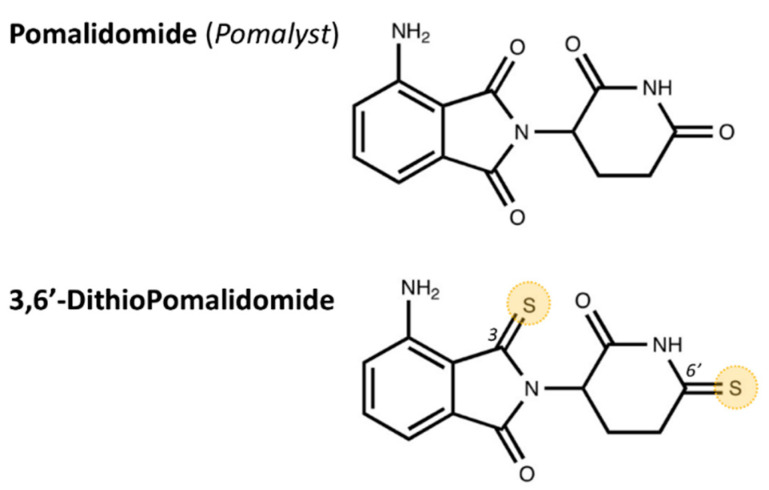
Chemical structures of Pomalidomide (Pom; clinically known as *Pomalyst*) and the new analog 3,6′-DithioPomalidomide (DP), with sulfurs shown in the marked 3 and 6′ positions (yellow highlights).

**Figure 2 ijms-22-08276-f002:**
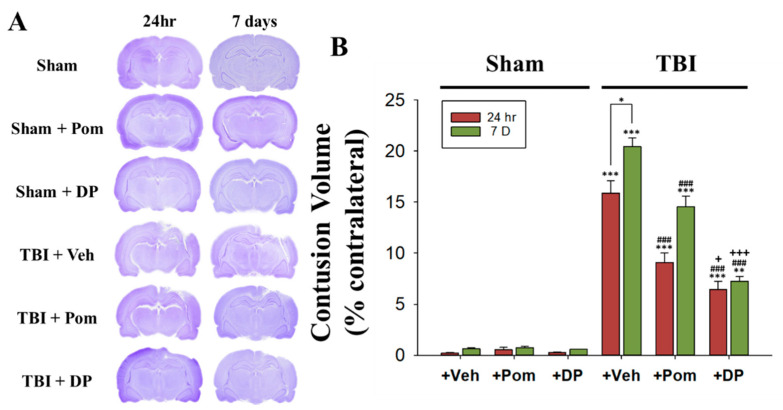
DP was more effective regarding the contusion volume reduction than Pom at 24 h and 7 days after a TBI. (**A**) Representative images of cresyl-violet-stained coronal brain sections from Sham + Veh, Sham + Pom, Sham + DP, TBI + Veh, TBI + Pom and TBI + DP rats at 24 h and 7 days after a TBI. (**B**) The contusion volume measured at 24 h and 7 days after a TBI was significantly reduced due to the administration of DP or Pom. Data are expressed as the mean ± S.E.M. * *p* < 0.05; ** *p* < 0.01; *** *p* < 0.001 versus the Sham + Veh group at the same time points versus different time points in the same group; ### *p* < 0.001 versus the TBI + Veh group at the same time point; + *p* < 0.05, +++ *p* < 0.001 versus the TBI + Pom group at the same time point (*n* = 5 in each group). Statistical differences were analyzed using independent sample t-tests and one-way ANOVA, followed by Tukey’s post hoc comparisons.

**Figure 3 ijms-22-08276-f003:**
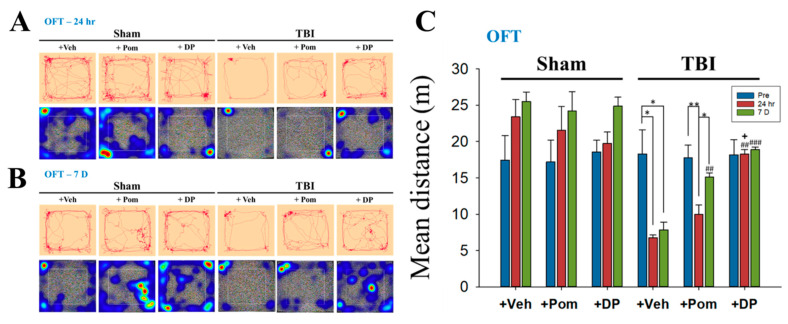
DP or Pom significantly improved TBI-induced anxiety-like behavior at 7 days, with the DP group showing the earlier onset of anxiety amelioration within 24 h after TBI. Representative traces and heatmaps of rat movements during the performance of OFT at 24 h (**A**) and 7 days (**B**) after TBI. On the heatmaps, the places where the rat stayed longer is shown in red. (**C**) The mean overall distance traveled at different time points in the Sham + Veh, Sham + Pom, Sham + DP, TBI + Veh, TBI + Pom and TBI + DP groups. Data are expressed as the mean ± SEM. * *p* < 0.05, ** *p* < 0.01 versus different time points in the same group; ## *p* < 0.01, ### *p* < 0.001 versus TBI + Veh group at the same time point; + *p* < 0.05 versus TBI + Pom group at the same time point (*n* = 5 in each group). Statistical differences were analyzed using one-way ANOVA, followed by Tukey’s post hoc comparisons.

**Figure 4 ijms-22-08276-f004:**
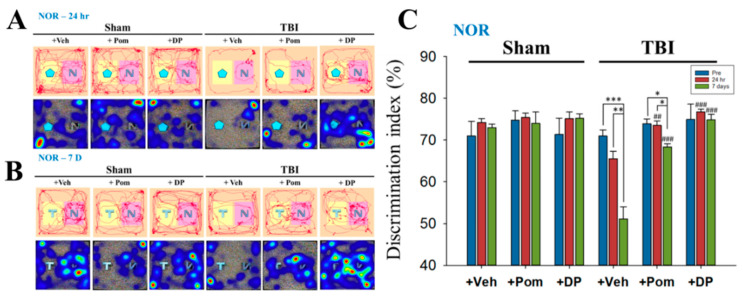
DP completely restored the TBI-induced short-term memory loss at 7D, whereas Pom only partially improved it when compared to pre-TBI levels. Representative traces and heatmaps of rat movements during the performance of NOR at 24 h (**A**) and 7 days (**B**) after a TBI. On the heatmaps, the place where the rat stayed longer is shown in red. (**C**) The discrimination index at different time points in the Sham + Veh, Sham + Pom, Sham + DP, TBI + Veh, TBI + Pom and TBI + DP groups. Data are expressed as the mean ± SEM. * *p* < 0.05, ** *p* < 0.01, *** *p* < 0.001 versus different time points in the same group; ## *p* < 0.01, ### *p* < 0.001 versus TBI + Veh group at the same time point (*n* = 5 in each group). Statistical differences were analyzed using one-way ANOVA, followed by Tukey’s post hoc comparisons.

**Figure 5 ijms-22-08276-f005:**
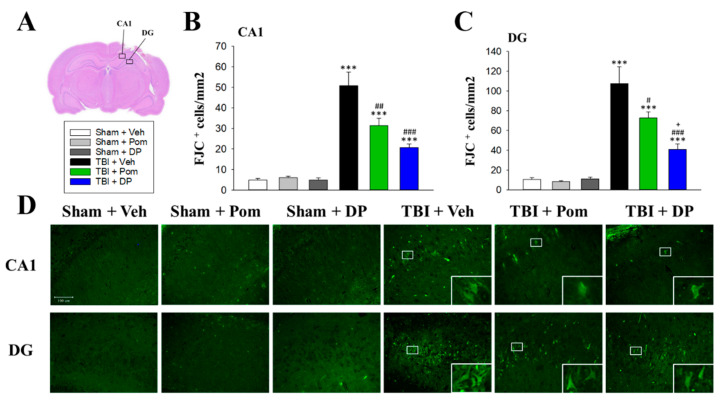
Post-injury administration of Pom or DP reduced degenerating neurons in the hippocampus 7 days after a TBI, with DP having better efficacy than Pom in the DG. (**A**) Representative image of an HE-stained coronal brain section from a TBI + Veh rat that shows the areas of evaluation at 7 days after a TBI. Quantitative comparison of mean densities of FJC+ cells in the hippocampal CA1 (**B**) and DG (**C**) regions at 7 days after a TBI. (**D**) Representative photomicrographs showing the FJC-stained hippocampal CA1 and DG regions from various groups at 7 days after a TBI. Data are expressed as the mean ± S.E.M. *** *p* < 0.001 versus Sham + Veh group; # *p* < 0.05, ## *p* < 0.01, ### *p* < 0.001 versus TBI + Veh group; + *p* < 0.05 versus TBI + Pom group (*n* = 5 in each group). Statistical difference was analyzed using independent sample t-tests and one-way ANOVA, followed by Tukey’s post hoc comparisons. Scale bar: 100 μm.

**Figure 6 ijms-22-08276-f006:**
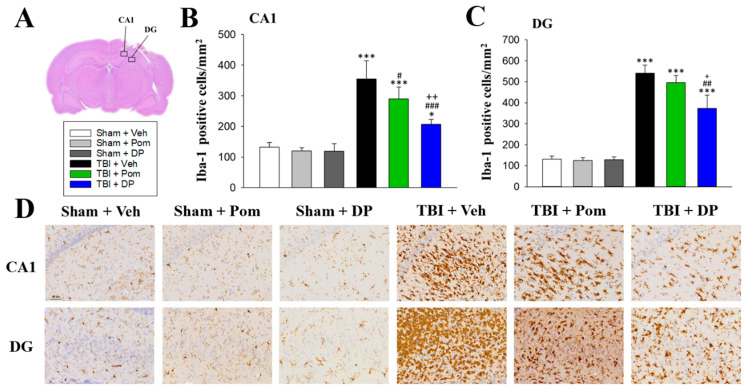
Only DP reduced the TBI-induced microgliosis in both hippocampus fields at 7D after a TBI. (**A**) Representative image of an HE-stained coronal brain section from TBI + Veh that shows the areas of evaluation at 7 days after a TBI. Quantitative comparison of mean densities of Iba-1-positive cells in the hippocampal CA1 (**B**) and DG (**C**) regions at 7 days after a TBI. (**D**) Representative photomicrographs showing the Iba-1-stained hippocampal CA1 and DG regions from various groups at 7 days after a TBI. Data are expressed as the mean ± S.E.M. * *p* < 0.05, *** *p* < 0.001 versus Sham + Veh group; # *p* < 0.05, ## *p* < 0.01, ### *p* < 0.001 versus TBI + Veh group; + *p* < 0.05, ++ *p* < 0.01 versus TBI + Pom group (*n* = 5 in each group). Statistical differences were analyzed using independent sample t-tests and one-way ANOVA, followed by Tukey’s post hoc comparisons. Scale bar: 60 μm.

**Figure 7 ijms-22-08276-f007:**
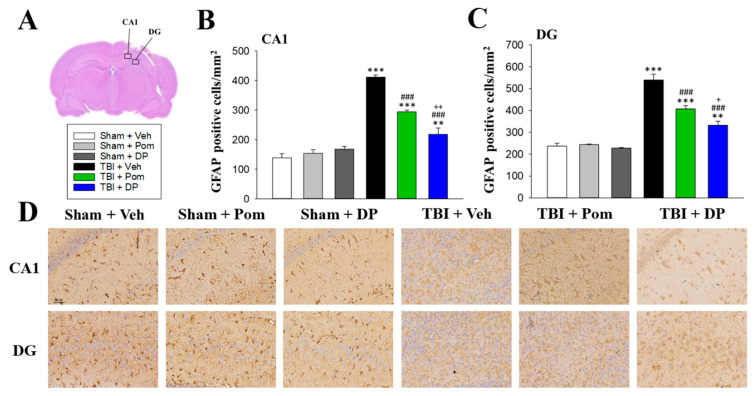
Post-injury administration of Pom or DP reduced TBI-induced astrogliosis in the hippocampus at 7 days after a TBI, with DP having better efficacy than Pom. (**A**) Representative image of an HE-stained coronal brain section from the TBI + Veh group that shows the areas of evaluation at 7 days after a TBI. Quantitative comparison of mean densities of GFAP positive cells in the hippocampal CA1 (**B**) and DG (**C**) regions at 7 days after a TBI. (**D**) Representative photomicrographs showing the GFAP-stained hippocampal CA1 and DG regions from various groups at 7 days after a TBI. Data are expressed as the mean ± S.E.M. ** *p* < 0.01, *** *p* < 0.001 versus Sham + Veh group; ### *p* < 0.001 versus TBI + Veh group; + *p* < 0.05, ++ *p* < 0.01 versus TBI + Pom group (*n* = 5 in each group). Statistical differences were analyzed using independent sample t-tests and one-way ANOVA, followed by Tukey’s post hoc comparisons. Scale bar: 60 μm.

## Data Availability

Not applicable.

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
