# Peer review of "3,6′-Dithiopomalidomide Ameliorates Hippocampal Neurodegeneration, Microgliosis and Astrogliosis and Improves Cognitive Behaviors in Rats with a Moderate Traumatic Brain Injury"

_ijms, 2021, doi:10.3390/ijms22158276_

Round 1
Reviewer 1 Report
Review of the manuscript entitled “3,6’-Dithiopomalidomide ameliorates hippocampal neurodegeneration, microgliosis, astrogliosis, and improves cognitive behaviors in rats with moderate traumatic brain injury”.
The manuscript is interesting but there are some aspects that need to be improved.
Introduction should be corrected. Exactly lines 92-112. Introduction must ended the aim of the paper. These lines are a summary of the work. It must be correct.
If everything is written in the introduction, why read the manuscript? In this lines should be information only about 3,6’-dithiopomalidomide (DP) and pomalidomide 92 (Pom) without Yours data (discoveries).
The quality of the figures, especially fig 1, should be improved. Line 224 to 228 should be moved to discussion or conclusion section.
The interpretation of the results, and even more so, the summary is unacceptable in the results chapter. Line 461 I think should be 50-60oC.
line 486-493, I think conclusion should be after discussion section. In general, the discussion needs to be corrected because the authors do not refer to their present results but discuss the work of other authors. The studied group of animals is very small, and there is no molecular explanation of the results obtained. The discussion is speculative.
Author Response
- Introduction should be corrected. Exactly lines 92-112. Introduction must ended the aim of the paper. These lines are a summary of the work. It must be correct. If everything is written in the introduction, why read the manuscript? In this lines should be information only about 3,6’-dithiopomalidomide (DP) and pomalidomide 92 (Pom) without Yours data (discoveries).
Response: The sentences in line 92-112 are corrected as follows:
The current study characterized the therapeutic potential of a novel pomalidomide (Pom) agent, 3,6’-dithiopomalidomide (DP) (Fig. 1), in comparison with Pom in the hippocampus following TBI induced by controlled cortical impact (CCI) in rat. DP and Pom are members of the immunomodulatory imide drug (IMiD) class, and lower TNF-α synthesis and hence down-stream signaling [25]. We previously reported favorable actions of DP [25] and Pom [25, 26] in CCI-induced TBI in cerebral cortex acutely, over the first 24 hr post injury period. We now report the activity of DP and Pom over a longer 7-day evaluation interval. This study involved documentation of the progressive neuropathological changes as well as efficacy with respect to contusion volume, neuronal cell death and functional outcome evaluated across a variety of behavioral and immunohistochemical parameters. The main objective of these studies was to evaluate longer-term progressive neuropathological changes in our CCI model of TBI and effects of DP and Pom on these changes. Both DP and Pom at the dose of 0.5 mg/kg (i.v.) were tolerated well by all animals, as appraised by wellbeing: evaluated from the subjective measures of grooming and appearance, righting skills, ambulation, and blinking reflex [27].
Our previous study demonstrated that 3,6’-dithiopomalidomide (DP), a novel analog of pomalidomide (Pom), can suppress TBI-induced astrogliosis and microgliosis in the cerebral cortex at 24 hr after TBI. In addition, DP can improve sensorimotor behavior outcomes following TBI [25]. In the present study, we investigated the effectiveness of Pom and DP on hippocampal injury, neuroinflammation and neurodegeneration at 7 days after TBI. Furthermore, to correlate our findings with clinical efficacy, we evaluated whether Pom and DP could improve TBI-induced cognitive deficits, including short-term memory impairment and anxiety-like behavior.
- The quality of the figures, especially fig 1, should be improved. Line 224 to 228 should be moved to discussion or conclusion section.
Response: The previous Fig. 1 is now replaced by a high resolution figure, and sentences in Line 224 to 228 are removed and now repositioned in the discussion section (and combined with old discussion text in Line 418-425).
Taken together, our results show progressive increases in TBI-induced cellular pathological processes reflected by increases in neuronal degeneration, microgliosis and astrogliosis within the hippocampus. In all cases, DP and, to a lesser degree, Pom reduced these 24 hr and 7 D changes, which were associated with parallel favorable changes in contusion volume, anxiety-like locomotor behavior, and short-term memory deficits induced by TBI.
- The interpretation of the results, and even more so, the summary is unacceptable in the results chapter. Line 461 I think should be 50-60oC.
Response: Thanks for these comments. We now removed the last paragraph beginning with “Taken together…..” at the end of the Results section (Line 224-228 in previous draft as suggested in previous comments) and placed them into discussion (with some revision, now Line 418-425 in the revised manuscript).
The typo in Line 461 is now corrected to 50-60℃ (Now Line 522)
- line 486-493, I think conclusion should be after discussion section. In general, the discussion needs to be corrected because the authors do not refer to their present results but discuss the work of other authors. The studied group of animals is very small, and there is no molecular explanation of the results obtained. The discussion is speculative.
Response: Thank you for these comments. We have now moved the conclusion to after the discussion section. (Lines 437-443)
We also rewrote the discussion and referred to our results following each Figure (from Fig. 2 to Fig. 7 with particular emphasis on cognitive behaviors and neuroinflammation) as well as deleted some discussion on the work of other authors.
We initially decided to use 5 rats for each group because it is a commonly used number to test for statistical significance when studying multiple groups. Only in the groups when 5 animals did not show a clear statistical trend, would we include more animals according to 3R principles (reduction, replacement and refinement) for ethical use of animals. A study conducted by Charan and Kantharia (2013) suggested that any sample size, which keeps E (E = Total number of animals − Total number of groups) between 10 and 20 should be considered as an adequate based on ANOVA tests. The E in our study is 24 (E=5*6-6) which is more than 20 indicating that adding more animals would not increase the chance of getting significant results.

Reviewer 2 Report
This article is great! I suggest you to amplify a little bit your bibliographic research adding these articles:
- Immunohistochemical Evaluation of Aquaporin-4 and its Correlation with CD68, IBA-1, HIF-1α, GFAP, and CD15 Expressions in Fatal Traumatic Brain Injury
- Traumatic brain injury: estimate of the age of the injury based on neuroinflammation, endothelial activation markers and adhesion molecules
Good Luck!
Author Response
This article is great! I suggest you to amplify a little bit your bibliographic research adding these articles:
- Immunohistochemical Evaluation of Aquaporin-4 and its Correlation with CD68, IBA-1, HIF-1α, GFAP, and CD15 Expressions in Fatal Traumatic Brain Injury
- Traumatic brain injury: estimate of the age of the injury based on neuroinflammation, endothelial activation markers and adhesion molecules
Good Luck!
Response: We thank Reviewer 2 for the comments and suggestions. We have added the suggested references (now numbered as 64, 65) and relevant text in the Discussion section (Discussion, lines 380-387 in the revised manuscript)

Reviewer 3 Report
The study design and experiments have been well-conducted and the authors did a great job in explaining their work. I have a few questions about the results for contusion volume. From figure 2 A: TBI+Pom image. Can the authors explain more why that small region in the injury area looks separated? While TBI+BP image looks better than TBI+Pom, the graphs show otherwise. Please comment or maybe add better representative images. Also at 24 h as well there seems to be a smaller injury area as compared to other conditions. How did the authors make sure the injury area was the same, to begin with in the TBI+DP condition as compared to the other conditions?
Image resolution is very poor for figure 1, the color also does not seem orange but yellow.
It will be beneficial to refer back to the main figures in the discussion to connect the argument with the authors’ results.
Author Response
- The study design and experiments have been well-conducted and the authors did a great job in explaining their work. I have a few questions about the results for contusion volume. From figure 2 A: TBI+Pom image. Can the authors explain more why that small region in the injury area looks separated? While TBI+BP image looks better than TBI+Pom, the graphs show otherwise. Please comment or maybe add better representative images. Also at 24 h as well there seems to be a smaller injury area as compared to other conditions. How did the authors make sure the injury area was the same, to begin with in the TBI+DP condition as compared to the other conditions?
Response: For Figure 2A, the small region in the injury area looking separated in the TBI + Pom group was due to an artifact in tissue sectioning. We did have better representative images from animals in this group so we replaced the image.
We used the identical impactor tip and experimental conditions to ensure that the degree of injury was the same in each animal. The details are as follows (originally written in the methods):
A 5-mm craniotomy is performed over the left parietal cortex, centered on the coronal suture and 3.5 mm lateral to the sagittal suture. The rat CCI model uses an electromagnetic impactor device that possessed a rounded (5-mm diameter) tip at a velocity of 4 m/s to a depth of 2 mm below the dura which resulted in an injury of moderate severity.
- Image resolution is very poor for figure 1, the color also does not seem orange but yellow.
Response: Thanks for your suggestion. The previous fig. 1 is replaced by a high resolution one and we changed the color description from orange to yellow.
- It will be beneficial to refer back to the main figures in the discussion to connect the argument with the authors’ results.
Response: We added figure numbers in the Discussion to refer back to the main figures (Fig. 2 to Fig. 7). In addition, we added the following discussion on the results cognitive behavior (Discussion, lines 329-344) and revised the discussion on neuroinflammation (Discussion, lines 373-412)

Reviewer 4 Report
Paper written by Pen-Sen Huang, Ping-Yen Tsai and colleagues is devoted to the investigation of therapeutic effect of novel compound 3,6’-Dithiopomalidomide (DP) in rat model of traumatic brain injury. Authors have clearly demonstrated the presence of neuroprotective effect of DP, observed the decrease in microgliosis and astrogliosis as well as improvment of behaviour in rodent model of TBI after single intrevenous administration of DP.
In the previous study (publishe in eLife) authors have demonstrated molecular mechanism of DP (lowering of TNF-α generation in plasma, hippocampus and ) and observed similar theraputic effects of DP in the cortex.
The current paper is a preclinical study of the DP-mediated anti-TBI therapeutic effect in the hippocampus.
Well structured paper with interesting results and excellent quality data, easy to follow.
Suggestion: accept in the present form
Author Response
Paper written by Pen-Sen Huang, Ping-Yen Tsai and colleagues is devoted to the investigation of therapeutic effect of novel compound 3,6’-Dithiopomalidomide (DP) in rat model of traumatic brain injury. Authors have clearly demonstrated the presence of neuroprotective effect of DP, observed the decrease in microgliosis and astrogliosis as well as improvement of behaviour in rodent model of TBI after single intravenous administration of DP.
In the previous study (published in eLife) authors have demonstrated molecular mechanism of DP (lowering of TNF-α generation in plasma, hippocampus and ) and observed similar therapeutic effects of DP in the cortex.
The current paper is a preclinical study of the DP-mediated anti-TBI therapeutic effect in the hippocampus.
Well structured paper with interesting results and excellent quality data, easy to follow.
Suggestion: accept in the present form
Response:
We appreciate the reviewer for the positive comment and the encouragement.
Round 2
Reviewer 1 Report
I believe that in its current form, the manuscript can be accepted
Author Response
We thank reviewer 1 for the approval of this manuscript. Those first-run comments are indeed very helpful to improve our manuscript.